# SENSITIVITY AND GENERALIZATION IN NEURAL NETWORKS: AN EMPIRICAL STUDY

**Roman Novak, Yasaman Bahri,**[*] **Daniel A. Abolafia,**
**Jeffrey Pennington, Jascha Sohl-Dickstein**

Google Brain

{romann, yasamanb, danabo, jpennin, jaschasd}@google.com

## ABSTRACT

In practice it is often found that large over-parameterized neural networks generalize better than their smaller counterparts, an observation that appears to conflict with classical notions of function complexity, which typically favor smaller models. In this work, we investigate this tension between complexity and generalization through an extensive empirical exploration of two natural metrics of complexity related to sensitivity to input perturbations. Our experiments survey thousands of models with various fully-connected architectures, optimizers, and other hyper-parameters, as well as four different image classification datasets.

We find that trained neural networks are more robust to input perturbations in the vicinity of the training data manifold, as measured by the norm of the input-output Jacobian of the network, and that it correlates well with generalization. We further establish that factors associated with poor generalization – such as full-batch training or using random labels – correspond to lower robustness, while factors associated with good generalization – such as data augmentation and ReLU non-linearities – give rise to more robust functions. Finally, we demonstrate how the input-output Jacobian norm can be predictive of generalization at the level of individual test points.

## 1 INTRODUCTION

The empirical success of deep learning has thus far eluded interpretation through existing lenses of computational complexity (Blum & Rivest, 1988), numerical optimization (Choromanska et al., 2015; Goodfellow & Vinyals, 2014; Dauphin et al., 2014) and classical statistical learning theory (Zhang et al., 2016): neural networks are highly non-convex models with extreme capacity that train fast and generalize well. In fact, not only do large networks demonstrate good test performance, but *larger* networks often generalize *better*, counter to what would be expected from classical measures, such as VC dimension. This phenomenon has been observed in targeted experiments (Neyshabur et al., 2015), historical trends of Deep Learning competitions (Canziani et al., 2016), and in the course of this work (Figure 1).

This observation is at odds with Occam's razor, the principle of parsimony, as applied to the intuitive notion of function complexity (see §A.2 for extended discussion). One resolution of the apparent contradiction is to examine complexity of functions in conjunction with the input domain. $f(x) = x^3 \sin(x)$ may seem decisively more complex than $g(x) = x$. But restrained to a narrow input domain of $[-0.01, 0.01]$ they appear differently: $g$ remains a linear function of the input, while $f(x) = \mathcal{O}\left(x^4\right)$ resembles a constant 0. In this work we find that such intuition applies to neural networks, that behave very differently close to the data manifold than away from it (§4.1).

We therefore analyze the complexity of models through their capacity to distinguish different inputs in the neighborhood of datapoints, or, in other words, their sensitivity. We study two simple metrics presented in §3 and find that one of them, the norm of the input-output Jacobian, correlates with generalization in a very wide variety of scenarios.

---

[*]Work done as a member of the Google Brain Residency program (g.co/brainresidency)

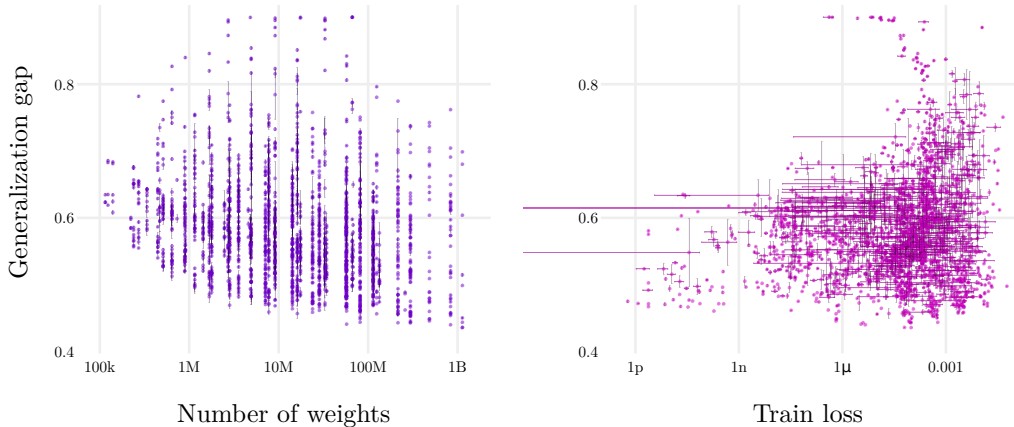

Figure 1: 2160 networks trained to 100% training accuracy on CIFAR10 (see §A.5.5 for experimental details). **Left**: while increasing capacity of the model allows for overfitting (top), very few models do, and a model with the maximum parameter count yields the best generalization (bottom right). **Right**: train loss does not correlate well with generalization, and the best model (minimum along the $y$-axis) has training loss many orders of magnitude higher than models that generalize worse (left). This observation rules out underfitting as the reason for poor generalization in low-capacity models. See (Neyshabur et al., 2015) for similar findings in the case of achievable 0 training loss.

This work considers sensitivity only in the context of image classification tasks. We interpret the observed correlation with generalization as an expression of a universal prior on (natural) image classification functions that favor robustness (see §A.2 for details). While we expect a similar prior to exist in many other perceptual settings, care should be taken when extrapolating our findings to tasks where such a prior may not be justified (e.g. weather forecasting).

## 1.1 PAPER OUTLINE

We first define sensitivity metrics for fully-connected neural networks in §3. We then relate them to generalization through a sequence of experiments of increasing level of nuance:

- In §4.1 we begin by comparing the sensitivity of trained neural networks on and off the training data manifold, i.e. in the regions of best and typical (over the whole input space) generalization.
- In §4.2 we compare sensitivity of identical trained networks that differ in a single hyper-parameter which is important for generalization.
- Further, §4.3 associates sensitivity and generalization in an unrestricted manner, i.e. comparing networks of a wide variety of hyper-parameters such as width, depth, non-linearity, weight initialization, optimizer, learning rate and batch size.
- Finally, §4.4 explores how predictive sensitivity (as measured by the Jacobian norm) is for individual test points.

## 1.2 SUMMARY OF CONTRIBUTIONS

The novelty of this work can be summarized as follows:

- Study of the behavior of trained neural networks on and off the data manifold through sensitivity metrics (§4.1).
- Evaluation of sensitivity metrics on trained neural networks in a very large-scale experimental setting and finding that they correlate with generalization (§4.2, §4.3, §4.4).

§2 puts our work in context of related research studying complexity, generalization, or sensitivity metrics similar to ours.

## 2 RELATED WORK

### 2.1 COMPLEXITY METRICS

We analyze complexity of fully-connected neural networks for the purpose of model comparison through the following sensitivity measures (see §3 for details):

- estimating the number of linear regions a network splits the input space into;
- measuring the norm of the input-output Jacobian within such regions.

A few prior works have examined measures related to the ones we consider. In particular, Pascanu et al. (2013); Montúfar et al. (2014); Raghu et al. (2016) have investigated the expressive power of fully-connected neural networks built out of piecewise-linear activation functions. Such functions are themselves piecewise-linear over their input domain, so that the number of linear regions into which input space is divided is one measure of how nonlinear the function is. A function with many linear regions has the capacity to build complex, flexible decision boundaries. It was argued in (Pascanu et al., 2013; Montúfar et al., 2014) that an upper bound to the number of linear regions scales exponentially with depth but polynomially in width, and a specific construction was examined. Raghu et al. (2016) derived a tight analytic bound and considered the number of linear regions for generic networks with random weights, as would be appropriate, for instance, at initialization. However, the evolution of this measure after training has not been investigated before. We examine a related measure, the number of hidden unit transitions along one-dimensional trajectories in input space, for trained networks. Further motivation for this measure is discussed in §3.

Another perspective on function complexity can be gained by studying their robustness to perturbations to the input. Indeed, Rasmussen & Ghahramani (2000) demonstrate on a toy problem how complexity as measured by the number of parameters may be of limited utility for model selection, while measuring the output variation allows the invocation of Occam's razor. In this work we apply related ideas to a large-scale practical context of neural networks with up to a billion free parameters (§4.2, §4.3) and discuss potential ways in which sensitivity permits the application of Occam's razor to neural networks (§A.2).

Sokolic et al. (2017) provide theoretical support for the relevance of robustness, as measured by the input-output Jacobian, to generalization. They derive bounds for the generalization gap in terms of the Jacobian norm within the framework of algorithmic robustness (Xu & Mannor, 2012). Our results provide empirical support for their conclusions through an extensive number of experiments. Several other recent papers have also focused on deriving tight generalization bounds for neural networks (Bartlett et al., 2017; Dziugaite & Roy, 2017; Neyshabur et al., 2018). We do not propose theoretical bounds in this paper but establish a correlation between our metrics and generalization in a substantially larger experimental setting than undertaken in prior works.

### 2.2 REGULARIZATION

In the context of regularization, increasing robustness to perturbations is a widely-used strategy: data augmentation, noise injection (Jiang et al., 2009), weight decay (Krogh & Hertz, 1992), and max-pooling all indirectly reduce sensitivity of the model to perturbations, while Rifai et al. (2011); Sokolic et al. (2017) explicitly penalize the Frobenius norm of the Jacobian in the training objective.

In this work we relate several of the above mentioned regularizing techniques to sensitivity, demonstrating through extensive experiments that improved generalization is consistently coupled with better robustness as measured by a single metric, the input-output Jacobian norm (§4.2). While some of these findings confirm common-sense expectations (random labels increase sensitivity, Figure 4, top row), others challenge our intuition of what makes a neural network robust (ReLU-networks, with unbounded activations, tend to be more robust than saturating HardSigmoid-networks, Figure 4, third row).

### 2.3 INDUCTIVE BIAS OF SGD

One of our findings demonstrates an inductive bias towards robustness in stochastic mini-batch optimization compared to full-batch training (Figure 4, bottom row). Interpreting this regularizing effect

in terms of some measure of sensitivity, such as curvature, is not new (Hochreiter & Schmidhuber, 1997; Keskar et al., 2016), yet we provide a novel perspective by relating it to reduced sensitivity to *inputs* instead of parameters.

The inductive bias of SGD ("implicit regularization") has been previously studied in (Neyshabur et al., 2015), where it was shown through rigorous experiments how increasing the width of a single-hidden-layer network improves generalization, and an analogy with matrix factorization was drawn to motivate constraining the norm of the weights instead of their count. Neyshabur et al. (2017) further explored several weight-norm based measures of complexity that do not scale with the size of the model. One of our measures, the Frobenius norm of the Jacobian is of similar nature (since the Jacobian matrix size is determined by the task and not by a particular network architecture). However, this particular metric was not considered, and, to the best of our knowledge, we are the first to evaluate it in a large-scale setting (e.g. our networks are up to 65 layers deep and up to $2^{16}$ units wide).

## 2.4 ADVERSARIAL EXAMPLES

Sensitivity to inputs has attracted a lot of interest in the context of adversarial examples (Szegedy et al., 2013). Several attacks locate points of poor generalization in the directions of high sensitivity of the network (Goodfellow et al., 2014; Papernot et al., 2016; Moosavi-Dezfooli et al., 2016), while certain defences regularize the model by penalizing sensitivity (Gu & Rigazio, 2014) or employing decaying (hence more robust) non-linearities (Kurakin et al., 2016). However, work on adversarial examples relates highly specific perturbations to a similarly specific kind of generalization (i.e. performance on a very small, adversarial subset of the data manifold), while this paper analyzes *average-case* sensitivity (§3) and *typical* generalization.

## 3 SENSITIVITY METRICS

We propose two simple measures of sensitivity for a fully-connected neural network (without biases) $\mathbf{f} : \mathbb{R}^d \to \mathbb{R}^n$ with respect to its input $\mathbf{x} \in \mathbb{R}^d$ (the output being unnormalized logits of the $n$ classes). Assume $\mathbf{f}$ employs a piecewise-linear activation function, like ReLU. Then $\mathbf{f}$ itself, as a composition of linear and piecewise-linear functions, is a piecewise-linear map, splitting the input space $\mathbb{R}^d$ into disjoint regions, implementing a single affine mapping on each. Then we can measure two aspects of sensitivity by answering

1. How does the output of the network change as the input is perturbed within the linear region?

2. How likely is the linear region to change in response to change in the input?

We quantify these qualities as follows:

1. For a local sensitivity measure we adopt the Frobenius norm of the class probabilities Jacobian $\mathbf{J}(\mathbf{x}) = \partial \mathbf{f}_\sigma(\mathbf{x}) / \partial \mathbf{x}^\mathbf{T}$ (with $J_{ij}(\mathbf{x}) = \partial [\mathbf{f}_\sigma(\mathbf{x})]_i / \partial x_j$), where $\mathbf{f}_\sigma = \sigma \circ \mathbf{f}$ with $\sigma$ being the softmax function[1]. Given points of interest $\mathbf{x}_\text{test}$, we estimate the sensitivity of the function in those regions with the average Jacobian norm:

$$\mathbb{E}_{\mathbf{x}_\text{test}} \left[ \| \mathbf{J}(\mathbf{x}_\text{test}) \|_F \right],$$

that we will further refer to as simply "Jacobian norm". Note that this does not require the labels for $\mathbf{x}_\text{test}$.

**Interpretation**. The Frobenius norm $\| \mathbf{J}(\mathbf{x}) \|_F = \sqrt{\sum_{ij} J_{ij}(\mathbf{x})^2}$ estimates the average-case sensitivity of $\mathbf{f}_\sigma$ around $\mathbf{x}$. Indeed, consider an infinitesimal Gaussian perturbation

---

[1] The norm of the Jacobian with respect to logits $\left( \partial \mathbf{f}(\mathbf{x}) / \partial \mathbf{x}^\mathbf{T} \right)$ experimentally turned out less predictive of test performance (not shown). See §A.3 for discussion of why the softmax Jacobian is related to generalization.

$\Delta \mathbf{x} \sim \mathcal{N}\left(\mathbf{0}, \epsilon \mathbf{I}\right)$: the expected magnitude of the output change is then

$$\mathbb{E}_{\Delta \mathbf{x}}\left[\left\|\mathbf{f}_{\sigma}\left(\mathbf{x}\right) - \mathbf{f}_{\sigma}\left(\mathbf{x} + \Delta \mathbf{x}\right)\right\|_2^2\right] \approx \mathbb{E}_{\Delta \mathbf{x}}\left[\left\|\mathbf{J}(\mathbf{x})\Delta \mathbf{x}\right\|_2^2\right] = \mathbb{E}_{\Delta \mathbf{x}}\Big[\sum_i \Big(\sum_j J_{ij} x_j\Big)^2\Big]$$

$$= \sum_{ijj'} J_{ij} J_{ij'} \mathbb{E}_{\Delta \mathbf{x}}\left[x_j x_{j'}\right] = \sum_{ij} J_{ij}^2 \mathbb{E}_{\Delta \mathbf{x}}\left[x_j^2\right]$$

$$= \epsilon \left\|\mathbf{J}\left(\mathbf{x}\right)\right\|_F^2 .$$

2. To detect a change in linear region (further called a "transition"), we need to be able to identify it. We do this analogously to Raghu et al. (2016). For a network with piecewise-linear activations, we can, given an input $\mathbf{x}$, assign a code to each neuron in the network $\mathbf{f}$, that identifies the linear region of the pre-activation of that neuron. E.g. each ReLU unit will have 0 or 1 assigned to it if the pre-activation value is less or greater than 0 respectively. Similarly, a ReLU6 unit (see definition in §A.4) will have a code of 0, 1, or 2 assigned, since it has 3 linear regions[2]. Then, a concatenation of codes of all neurons in the network (denoted by $\mathbf{c}(\mathbf{x})$) uniquely identifies the linear region of the input $\mathbf{x}$ (see §A.1.1 for discussion of edge cases).

Given this encoding scheme, we can detect a transition by detecting a change in the code. We then sample $k$ equidistant points $\mathbf{z}_0, \ldots, \mathbf{z}_{k-1}$ on a closed one-dimensional trajectory $\mathcal{T}(\mathbf{x})$ (generated from a data point $\mathbf{x}$ and lying close to the data manifold; see below for details) and count transitions $t(\mathbf{x})$ along it to quantify the number of linear regions:

$$t(\mathbf{x}) := \sum_{i=0}^{k-1} \left\|\mathbf{c}\left(\mathbf{z}_i\right) - \mathbf{c}\left(\mathbf{z}_{(i+1)\%k}\right)\right\|_1 \approx \oint_{\mathbf{z} \in \mathcal{T}(\mathbf{x})} \left\|\frac{\partial \mathbf{c}(\mathbf{z})}{\partial \left(d\mathbf{z}\right)}\right\|_1 d\mathbf{z}, \qquad (1)$$

where the norm of the directional derivative $\left\|\partial \mathbf{c}(\mathbf{z}) / \partial \left(d\mathbf{z}\right)\right\|_1$ amounts to a Dirac delta function at each transition (see §A.1.2 for further details).

By sampling multiple such trajectories around different points, we estimate the sensitivity metric:

$$\mathbb{E}_{\mathbf{x}_{\text{test}}}\left[t\left(\mathbf{x}_{\text{test}}\right)\right],$$

that we will further refer to as simply "transitions" or "number of transitions."

To assure the sampling trajectory $\mathcal{T}(\mathbf{x}_{\text{test}})$ is close to the data manifold (since this is the region of interest), we construct it through horizontal translations of the image $\mathbf{x}_{\text{test}}$ in pixel space (Figure App.7, right). We similarly augment our training data with horizontal and vertical translations in the corresponding experiments (Figure 4, second row).

As earlier, this metric does not require knowing the label of $\mathbf{x}_{\text{test}}$.

**Interpretation.** We can draw a qualitative parallel between the number of transitions and curvature of the function. One measure of curvature of a function $\mathbf{f}$ is the total norm of the directional derivative of its first derivative $\mathbf{f}'$ along a path:

$$C\left(\mathbf{f}, \mathcal{T}\left(\mathbf{x}\right)\right) := \oint_{\mathbf{z} \in \mathcal{T}(\mathbf{x})} \left\|\frac{\partial \mathbf{f}'\left(\mathbf{z}\right)}{\partial \left(d\mathbf{z}\right)}\right\|_F d\mathbf{z}.$$

A piecewise-linear function $\mathbf{f}$ has a constant first derivative $\mathbf{f}'$ everywhere except for the transition boundaries. Therefore, for a sufficiently large $k$, curvature can be expressed as

$$C\left(\mathbf{f}, \mathcal{T}\left(\mathbf{x}\right)\right) = \frac{1}{2} \sum_{i=0}^{k-1} \left\|\mathbf{f}'\left(\mathbf{z}_i\right) - \mathbf{f}'\left(\mathbf{z}_{(i+1)\%k}\right)\right\|_F,$$

where $\mathbf{z}_0, \ldots, \mathbf{z}_{k-1}$ are equidistant samples on $\mathcal{T}\left(\mathbf{x}\right)$. This sum is similar to $t(\mathbf{x})$ as defined in Equation 1, but quantifies the amount of change in between two linear regions in a non-binary way. However, estimating it on a densely sampled trajectory is a computationally-intensive task, which is one reason we instead count transitions.

---

[2] For a non-piecewise-linear activation like Tanh, we consider 0 as the boundary of two regions and find this metric qualitatively similar to counting transitions of a piecewise-linear non-linearity.

As such, on a qualitative level, the two metrics (Jacobian norm and number of transitions) track the first and second order terms of the Taylor expansion of the function.

Above we have defined two sensitivity metrics to describe the learned function around the data, on average. In §4.1 we analyze these measures on and off the data manifold by simply measuring them along circular trajectories in input space that intersect the data manifold at certain points, but generally lie away from it (Figure 2, left).

## 4 EXPERIMENTAL RESULTS

In the following subsections (§4.2, §4.3) each study analyzes performance of a large number (usually thousands) of fully-connected neural networks having different hyper-parameters and optimization procedures. Except where specified, we include only models which achieve $100\%$ training accuracy. This allows us to study generalization disentangled from properties like expressivity and trainability, which are outside the scope of this work.

In order to efficiently evaluate the compute-intensive metrics (§3) in a very wide range of hyper-parameters settings (see e.g. §A.5.5) we only consider fully-connected networks. Extending the investigation to more complex architectures is left for future work.

### 4.1 SENSITIVITY ON AND OFF THE TRAINING DATA MANIFOLD

We analyze the behavior of a trained neural network near and away from training data. We do this by comparing sensitivity of the function along 3 types of trajectories:

1.  A random ellipse. This trajectory is extremely unlikely to pass anywhere near the real data, and indicates how the function behaves in random locations of the input space that it never encountered during training.

2.  An ellipse passing through three training points of different class (Figure 2, left). This trajectory does pass through the three data points, but in between it traverses images that are linear combinations of different-class images, and are expected to lie outside of the natural image space. Sensitivity of the function along this trajectory allows comparison of its behavior on and off the data manifold, as it approaches and moves away from the three anchor points.

3.  An ellipse through three training points of the same class. This trajectory is similar to the previous one, but, given the dataset used in the experiment (MNIST), is expected to traverse overall closer to the data manifold, since linear combinations of the same digit are more likely to resemble a realistic image. Comparing transition density along this trajectory to the one through points of different classes allows further assessment of how sensitivity changes in response to approaching the data manifold.

We find that, according to both the Jacobian norm and transitions metrics, functions exhibit much more robust behavior around the training data (Figure 2, center and right). We further visualize this effect in 2D in Figure 3, where we plot the transition boundaries of the last (pre-logit) layer of a neural network before and after training. After training we observe that training points lie in regions of low transition density.

The observed contrast between the neural network behavior near and away from data further strengthens the empirical connection we draw between sensitivity and generalization in §4.2, §4.3 and §4.4; it also confirms that, as mentioned in §1, if a certain quality of a function is to be used for model comparison, input domain should always be accounted for.

### 4.2 SENSITIVITY AND GENERALIZATION FACTORS

In §4.1 we established that neural networks implement more robust functions in the vicinity of the training data manifold than away from it.

We now consider the more practical context of model selection. Given two perfectly trained neural networks, does the model with better generalization implement a less sensitive function?

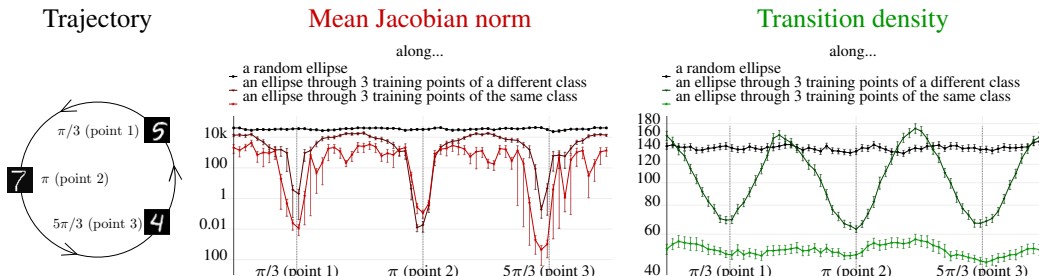

Figure 2: A 100%-accurate (on training data) MNIST network implements a function that is much more stable near training data than away from it. **Left**: depiction of a hypothetical circular trajectory in input space passing through three digits of different classes, highlighting the training point locations ($\pi/3, \pi, 5\pi/3$). **Center**: Jacobian norm as the input traverses an elliptical trajectory. Sensitivity drops significantly in the vicinity of training data while remaining uniform along random ellipses. **Right**: transition density behaves analogously. According to both metrics, as the input moves between points of different classes, the function becomes less stable than when it moves between points of the same class. This is consistent with the intuition that linear combinations of different digits lie further from the data manifold than those of same-class digits (which need not hold for more complex datasets). See §A.5.2 for experimental details.

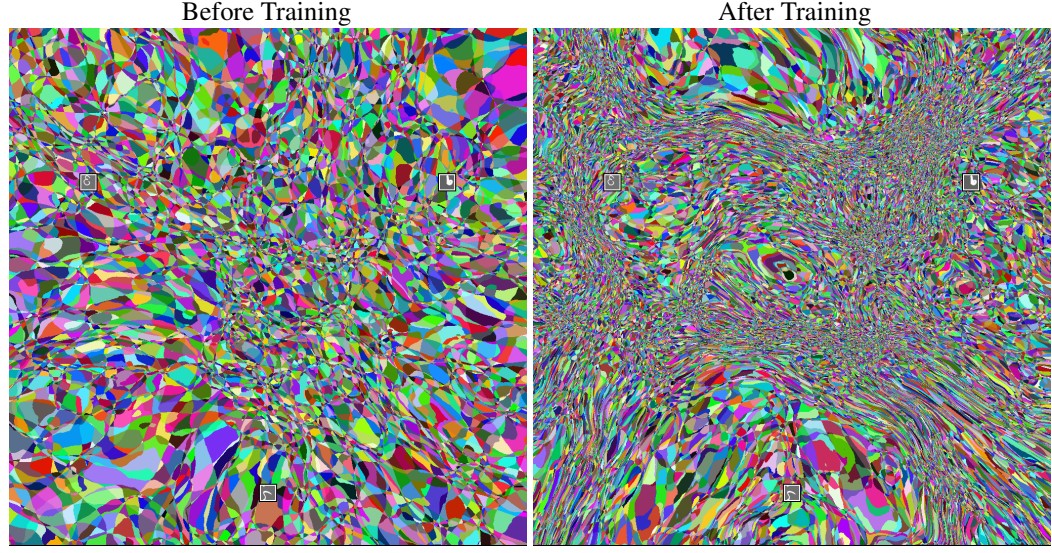

Figure 3: Transition boundaries of the last (pre-logits) layer over a 2-dimensional slice through the input space defined by 3 training points (indicated by inset squares). **Left**: boundaries before training. **Right**: after training, transition boundaries become highly non-isotropic, with training points lying in regions of lower transition density. See §A.5.3 for experimental details.

We study approaches in the machine learning community that are commonly believed to influence generalization (Figure 4, top to bottom):

- random labels;
- data augmentation;
- ReLUs;
- full-batch training.

We find that in each case, the change in generalization is coupled with the respective change in sensitivity (i.e. lower sensitivity corresponds to smaller generalization gap) as measured by the Jacobian norm (and almost always for the transitions metric).

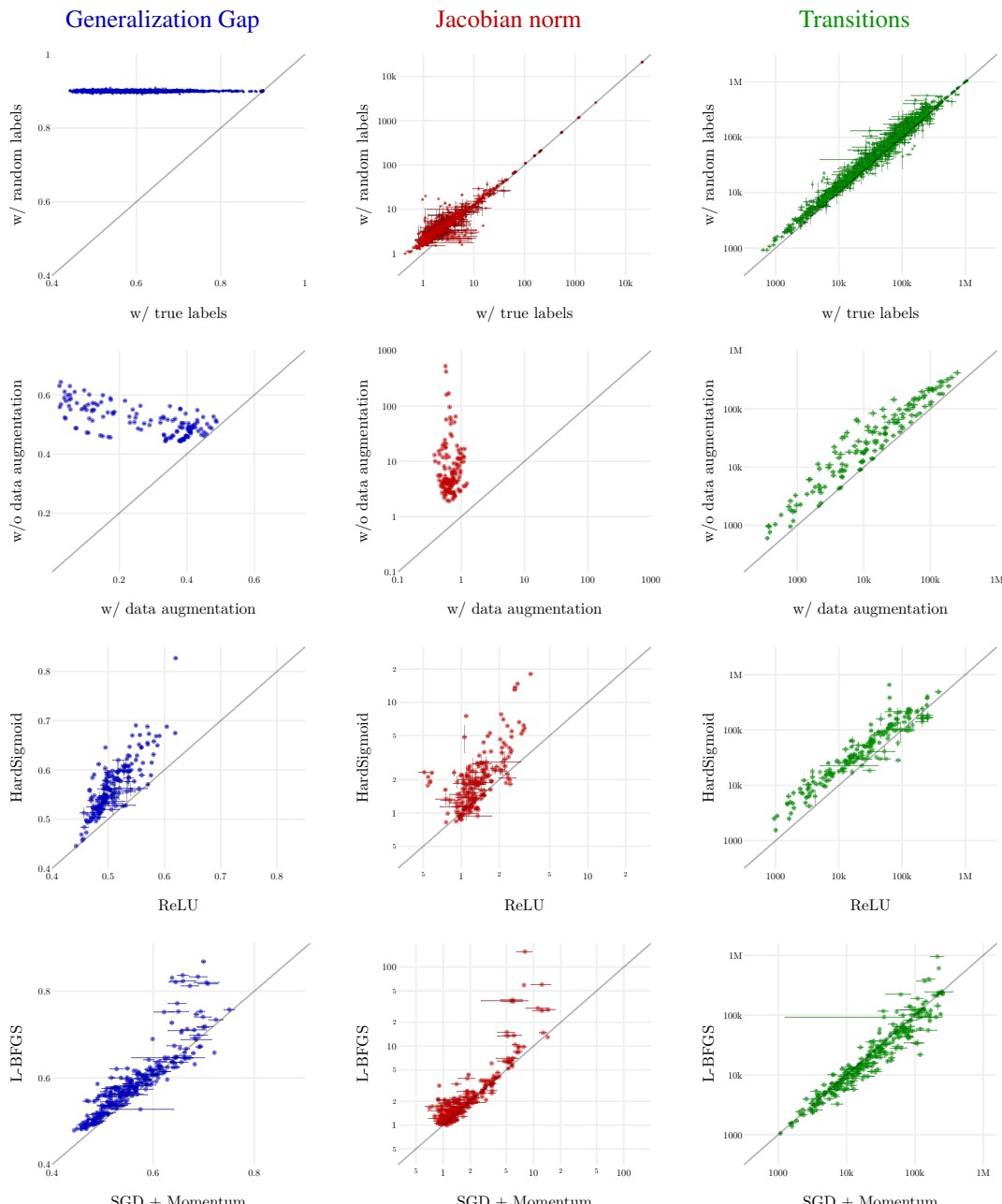

Figure 4: Improvement in generalization (left column) due to using correct labels, data augmentation, ReLUs, mini-batch optimization (top to bottom) is consistently coupled with reduced sensitivity as measured by the Jacobian norm (center column). Transitions (right column) correlate with generalization in all considered scenarios except for comparing optimizers (bottom right). Each point on the plot corresponds to two neural networks that share all hyper-parameters and the same optimization procedure, but differ in a certain property as indicated by axes titles. The coordinates along each axis reflect the values of the quantity in the title of the plot in the respective setting (i.e. with true or random labels). All networks have reached $100\%$ training accuracy on CIFAR10 in both settings (except for the data-augmentation study, second row; see §A.5.4 for details). See §A.5.5 for experimental details (§A.5.4 for the data-augmentation study) and §4.2.1 for plot interpretation.

### 4.2.1 HOW TO READ PLOTS

In Figure 4, for many possible hyper-parameter configurations, we train two models that share all parameters and optimization procedure, but differ in a single binary setting (i.e. trained on true or random labels; with or without data augmentation; etc). Out of all such network pairs, we select only those where each network reached 100% training accuracy on the whole training set (apart from the data augmentation study). The two generalization or sensitivity values are then used as the $x$ and $y$ coordinates of a point corresponding to this pair of networks (with the plot axes labels denoting the respective value of the binary parameter considered). The position of the point with respect to the diagonal $y = x$ visually demonstrates which configuration has smaller generalization gap / lower sensitivity.

### 4.3 SENSITIVITY AND GENERALIZATION GAP

We now perform a large-scale experiment to establish direct relationships between sensitivity and generalization in a realistic setting. In contrast to §4.1, where we selected locations in the input space, and §4.2, where we varied a single binary parameter impacting generalization, we now sweep simultaneously over many different architectural and optimization choices (§A.5.5).

Our main result is presented in Figure 5, indicating a strong relationship between the Jacobian norm and generalization. In contrast, Figure App.8 demonstrates that the number of transitions is not alone sufficient to compare networks of different sizes, as the number of neurons in the networks has a strong influence on transition count.

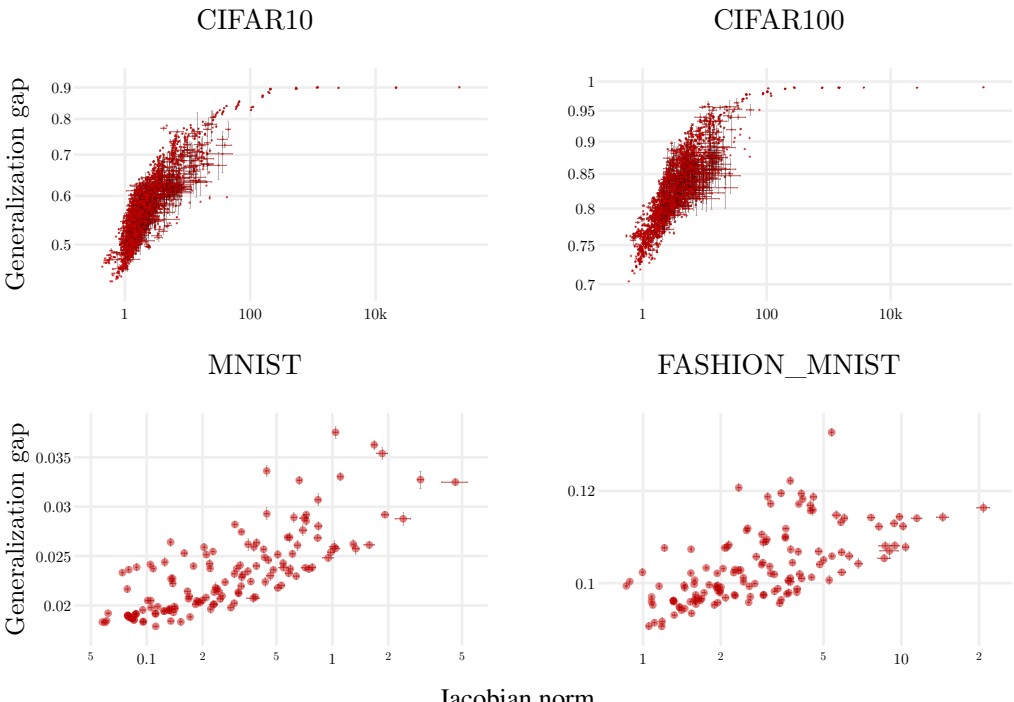

Figure 5: Jacobian norm correlates with generalization gap on all considered datasets. Each point corresponds to a network trained to 100% training accuracy (or at least 99.9% in the case of CIFAR100). See §A.5.4 and §A.5.5 for experimental details of bottom and top plots respectively.

## 4.4 SENSITIVITY AND PER-POINT GENERALIZATION

In §4.3 we established a correlation between sensitivity (as measured by the Jacobian norm) and generalization averaged over a large test set ($10^4$ points). We now investigate whether the Jacobian norm can be predictive of generalization at individual points.

As demonstrated in Figure 6 (top), Jacobian norm at a point is predictive of the cross-entropy loss, but the relationship is not a linear one, and not even bijective (see §A.3 for analytic expressions explaining it). In particular, certain misclassified points (right sides of the plots) have a Jacobian norm many orders of magnitude smaller than that of the correctly classified points (left sides). However, we do remark a consistent tendency for points having the highest values of the Jacobian norm to be mostly misclassified. A similar yet noisier trend is observed in networks trained using $\ell_2$-loss as depicted in Figure 6 (bottom). These observations make the Jacobian norm a promising quantity to consider in the contexts of active learning and confidence estimation in future research.

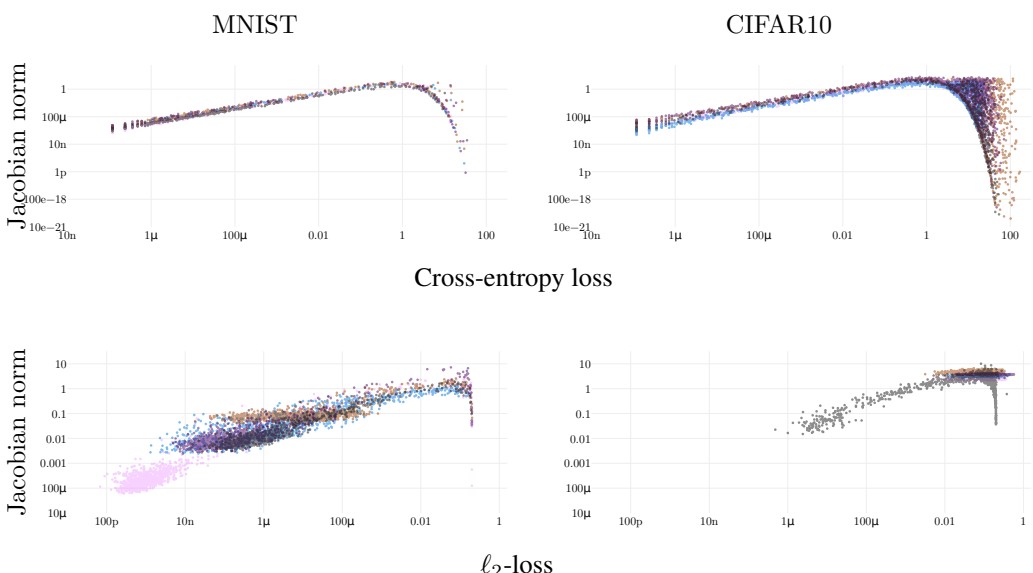

Figure 6: Jacobian norm plotted against individual test point loss. Each plot shows 5 random networks that fit the respective training set with $100\%$ accuracy, with each network having a unique color. **Top:** Jacobian norm plotted against cross-entropy loss. These plots experimentally confirm the relationship established in §A.3 and Figure App.11. **Bottom:** Jacobian norm plotted against $\ell_2$-loss, for networks trained on $\ell_2$-loss, exhibits a similar behavior. See §A.5.6 for experimental details and Figure App.9 for similar observations on other datasets.

## 5 CONCLUSION

We have investigated sensitivity of trained neural networks through the input-output Jacobian norm and linear regions counting in the context of image classification tasks. We have presented extensive experimental evidence indicating that the local geometry of the trained function as captured by the input-output Jacobian can be predictive of generalization in many different contexts, and that it varies drastically depending on how close to the training data manifold the function is evaluated. We further established a connection between the cross-entropy loss and the Jacobian norm, indicating that it can remain informative of generalization even at the level of individual test points. Interesting directions for future work include extending our investigation to more complex architectures and other machine learning tasks.

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

## A  APPENDIX

### A.1  TRANSITION METRIC DETAILS

#### A.1.1  LINEAR REGION ENCODING

The way of encoding a linear region $\mathbf{c}(\mathbf{z})$ of a point $\mathbf{z}$ described in §3 (2) guarantees that different regions obtain different codes, but different codes might be assigned to the same region if all the neurons in any layer of the network are saturated (or if weights leading from the transitioning unit to active units are exactly zero, or exactly cancel). However, the probability of such an arrangement drops exponentially with width and hence is ignored in this work.

#### A.1.2  TRANSITION COUNTING

The equality between the discrete and continuous versions of $t(\mathbf{x})$ in Equation 1 becomes exact with a high-enough sampling density $k$ such that there are no narrow linear regions missed in between consecutive points (precisely, the encoding $\mathbf{c}(\mathbf{z})$ has to only change at most once on the line between two consecutive points $\mathbf{z}_i$ and $\mathbf{z}_{i+1}$).

For computational efficiency we also assume that no two neurons transitions simultaneously, which is extremely unlikely in the context of random initialization and stochastic optimization.

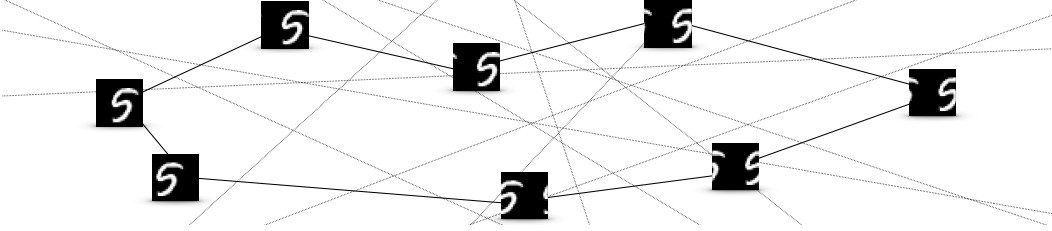

Figure App.7: Depiction of a trajectory in input space used to count transitions as defined in §3 (2). An interpolation between 28 horizontal translations of a single digit results in a complex trajectory that constrains all points to lie close to the translation-augmented data, and allows for a tractable estimate of transition density around the data manifold. This metric is used to compare models in §4.2 and §4.3. Straight lines indicate boundaries between different linear regions (straight-line boundaries between linear regions is accurate for the case of a single-layer piecewise-linear network. The partition into linear regions is more complex for deeper networks (Raghu et al., 2016)).

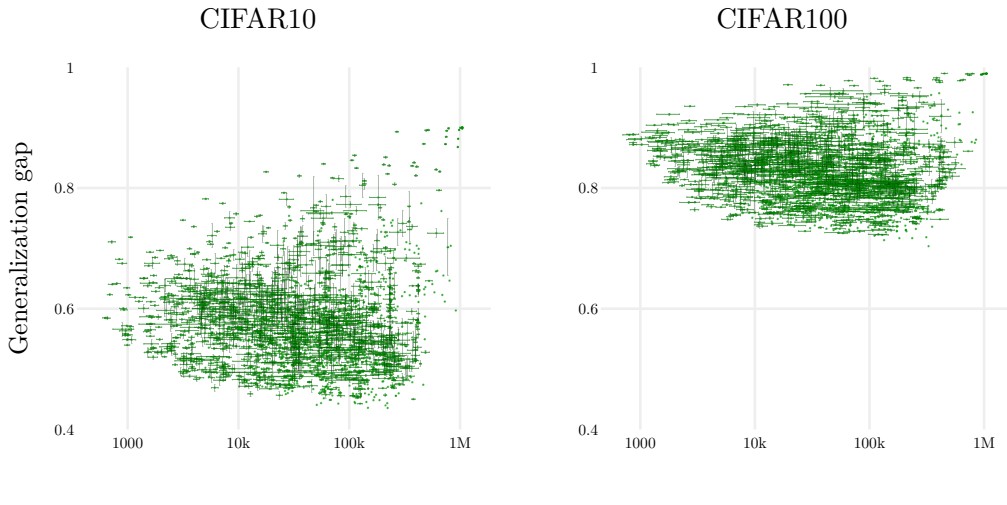

Figure App.8: Number of transitions, in contrast to Figure 5, does not generally correlate with generalization gap. **Left**: 2160 networks with 100% train accuracy on CIFAR10. **Right**: 2097 networks with at least 99.9% training accuracy on CIFAR100. See §A.5.5 for experimental details.

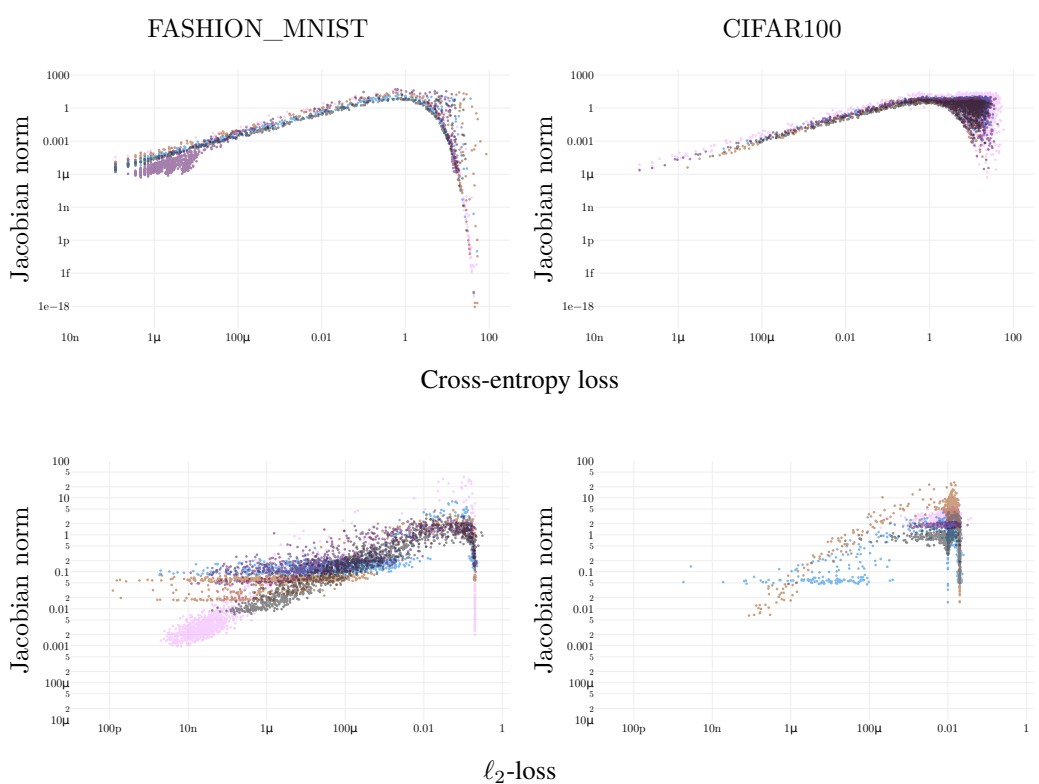

Figure App.9: Jacobian norm plotted against individual test point loss on Fashion-MNIST (Xiao et al., 2017) and CIFAR100. As in Figure 6, each plot shows 5 random networks that fit the respective training set to a 100% with each network having a unique color. See §A.5.6 for experimental details.

## A.2 Do NEURAL NETWORKS DEFY OCCAM'S RAZOR?

Here we briefly discuss the motivation of this work in the context of Occam's razor.

**Occam's razor** is a heuristic for model comparison based on their complexity. Given a dataset $\mathcal{D}$, Occam's razor gives preference to simpler models $\mathcal{H}$. In the Bayesian interpretation of the heuristic (Jefferys & Berger, 1992) simplicity is defined as evidence $\mathbb{P}[\mathcal{D}|\mathcal{H}]$ and is often computed using the Laplace approximation. Under further assumptions (MacKay, 1991), this evidence can be shown to be inversely proportional to the number of parameters in the model. Therefore, given a uniform prior $\mathbb{P}[\mathcal{H}]$ on two competing hypothesis classes, the class posterior $\mathbb{P}[\mathcal{H}|\mathcal{D}] \sim \mathbb{P}[\mathcal{D}|\mathcal{H}]\mathbb{P}[\mathcal{H}]$ is higher for a model with fewer parameters.

An alternative, qualitative justification of the heuristic is through considering the evidence as a normalized probability distribution over the whole dataset space:

$$\int_{\mathcal{D}'} \mathbb{P}[\mathcal{D}'|\mathcal{H}] \, d\mathcal{D}' = 1$$

and remarking that models with more parameters have to spread the probability mass more evenly across all the datasets by virtue of being able to fit more of them (Figure App.10, left). This similarly suggests (under a uniform prior on competing hypothesis classes) preferring models with fewer parameters, assuming that evidence is unimodal and peaks close to the dataset of interest.

**Occam's razor for neural networks.** As seen in Figure 1, the above reasoning does not apply to neural networks: the best achieved generalization is obtained by a model that has around $10^4$ times as many parameters as the simplest model capable of fitting the dataset (within the evaluated search space).

On one hand, Murray & Ghahramani (2005); Telgarsky (2015) demonstrate on concrete examples that a high number of free parameters in the model doesn't necessarily entail high complexity. On the other hand, a large body of work on the expressivity of neural networks (Pascanu et al., 2013; Montúfar et al., 2014; Raghu et al., 2016; Poole et al., 2016) shows that their ability to compute complex functions increases rapidly with size, while Zhang et al. (2016) validates that they also easily fit complex (even random) functions with stochastic optimization. Classical metrics like VC dimension or Rademacher complexity increase with size of the network as well. This indicates that weights of a neural network may actually correspond to its usable capacity, and hence "smear" the evidence $\mathbb{P}[\mathcal{D}|\mathcal{H}]$ along a very large space of datasets $\mathcal{D}'$, making the dataset of interest $\mathcal{D}$ less likely.

**Potential issues.** We conjecture the Laplace approximation of the evidence $\mathbb{P}[\mathcal{D}|\mathcal{H}]$ and the simplified estimation of the "Occam's factor" in terms of the accessible volume of the parameter space might not hold for neural networks in the context of stochastic optimization, and, in particular, do not account for the combinatorial growth of the accessible volume of parameter space as width increases (MacKay, 1992). Similarly, when comparing evidence as probability distributions over datasets, the difference between two neural networks may not be as drastic as in Figure App.10 (left), but more nuanced as depicted in Figure App.10 (right), with the evidence ratio being highly dependent on the particular dataset.

We interpret our work as defining hypothesis classes based on sensitivity of the hypothesis (which yielded promising results in (Rasmussen & Ghahramani, 2000) on a toy task) and observing a strongly non-uniform prior on these classes that enables model comparison. Indeed, at least in the context of natural images classification, putting a prior on the number of parameters or Kolmogorov complexity of the hypothesis is extremely difficult. However, a statement that the true classification function is robust to small perturbations in the input is much easier to justify. As such, a prior $\mathbb{P}[\mathcal{H}]$ in favor of robustness over sensitivity might fare better than a prior on specific network hyper-parameters.

Above is one way to interpret the correlation between sensitivity and generalization that we observe in this work. It does not explain why large networks tend to converge to less sensitive functions. We conjecture large networks to have access to a larger space of robust solutions due to solving a highly-underdetermined system when fitting a dataset, while small models converge to more extreme weight values due to being overconstrained by the data. However, further investigation is needed to confirm this hypothesis.

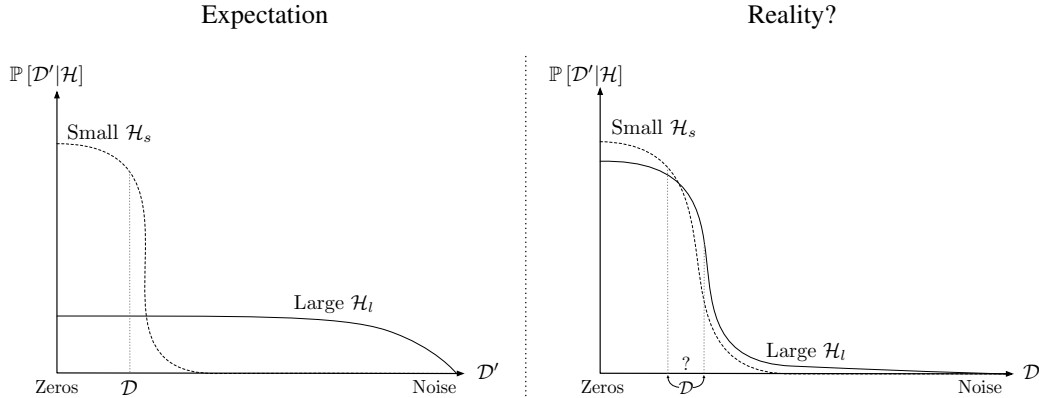

Figure App.10: Occam's razor: simplified expectation vs hypothesized reality. All datasets $\mathcal{D}'$ with input and target dimensions matching those of a particular dataset $\mathcal{D}$ are sorted according to the evidence $\mathbb{P}\left[\mathcal{D}'|\mathcal{H}\right]$ of a large model $\mathcal{H}_l$ from left to right along the horizontal axis. **Left**: a classic simplified depiction of Bayesian Occam's razor. Evidence $\mathbb{P}\left[\mathcal{D}'|\mathcal{H}\right]$ of a small model $\mathcal{H}_s$ with few parameters has narrow support in the dataset space and is more peaked. If the model fits the dataset $\mathcal{D}$ well, it falls close to the peak and outperforms a larger model $\mathcal{H}_l$ with more parameters, having wider support. **Right**: suggested potential reality of neural networks. Evidence of the small model $\mathcal{H}_s$ peaks higher, but the large model $\mathcal{H}_l$ might nonetheless concentrate the majority of probability mass on simple functions and the evidence curves might intersect at a small angle. In this case, while a dataset $\mathcal{D}$ lying close to the intersection can be fit by both models, the Bayesian evidence ratio depends on its exact position with respect to the intersection.

## A.3 Bounding the Jacobian Norm

Here we analyze the relationship between the Jacobian norm and the cross-entropy loss at individual test points as studied in §4.4.

**Target class Jacobian**. We begin by relating the derivative of the target class probability $\mathbf{J}_{y(\mathbf{x})}$ to per-point cross-entropy loss $l(\mathbf{x}) = -\log\left[\mathbf{f}_\sigma(\mathbf{x})\right]_{y(\mathbf{x})}$ (where $y(\mathbf{x})$ is the correct integer class).

We will denote $\mathbf{f}_\sigma(\mathbf{x})$ by $\boldsymbol{\sigma}$ and drop the $\mathbf{x}$ argument to de-clutter notation (i.e. write $\mathbf{f}$ instead of $\mathbf{f}(\mathbf{x})$). Then the Jacobian can be expressed as

$$\mathbf{J} = \left[\left(\boldsymbol{\sigma}\mathbf{1}^T\right) \odot \left(\mathbf{I} - \boldsymbol{\sigma}\mathbf{1}^T\right)^T\right]\left(\frac{\partial\mathbf{f}}{\partial\mathbf{x}^T}\right),$$

where $\odot$ is the Hadamard element-wise product. Then indexing both sides of the equation at the correct class $y$ yields

$$\mathbf{J}_y = \sigma_y\left(\left(\mathbf{e}_y - \boldsymbol{\sigma}\right)^T\left(\frac{\partial\mathbf{f}}{\partial\mathbf{x}^T}\right)\right),$$

where $\mathbf{e}_y$ is a vector of zeros everywhere except for $e_y = 1$. Taking the norm of both sides results in

$$\|\mathbf{J}_y\|_2^2 = \sigma_y^2\sum_{k=1}^d\left[(1-\sigma_y)^2\left(\frac{\partial f_y}{\partial x_k}\right)^2 + \sum_{j\neq y}^n\left(\sigma_j\frac{\partial f_j}{\partial x_k}\right)^2\right] \tag{2}$$

$$= \sigma_y^2\left[(1-\sigma_y)^2\sum_{k=1}^d\left(\frac{\partial f_y}{\partial x_k}\right)^2 + \sum_{j\neq y}^n\sigma_j^2\sum_{k=1}^d\left(\frac{\partial f_j}{\partial x_k}\right)^2\right] \tag{3}$$

$$= \sigma_y^2\left[(1-\sigma_y)^2\left\|\frac{\partial f_y}{\partial\mathbf{x}^T}\right\|_2^2 + \sum_{j\neq y}^n\sigma_j^2\left\|\frac{\partial f_j}{\partial\mathbf{x}^T}\right\|_2^2\right] \tag{4}$$

We now assume that magnitudes of the individual logit derivatives vary little in between logits and over the input space[3]:

$$\left\| \frac{\partial f_i}{\partial \mathbf{x}^T} \right\|_2^2 \approx \frac{1}{n} \mathbb{E}_{\mathbf{x}_{\text{test}}} \left\| \frac{\partial \mathbf{f}}{\partial \mathbf{x}_{\text{test}}^T} \right\|_F^2 ,$$

which simplifies Equation 4 to

$$\| \mathbf{J}_y \|_2^2 \approx M \sigma_y^2 \left[ (1 - \sigma_y)^2 + \sum_{j \neq y}^n \sigma_j^2 \right],$$

where $M = \mathbb{E}_{\mathbf{x}_{\text{test}}} \left\| \partial \mathbf{f} / \partial \mathbf{x}_{\text{test}}^T \right\|_F^2 / n$. Since $\sigma$ lies on the $(n-1)$-simplex $\Delta^{n-1}$, under these assumptions we can bound:

$$\frac{(1 - \sigma_y)^2}{n - 1} \leqslant \sum_{j \neq y}^n \sigma_j^2 \leqslant (1 - \sigma_y)^2,$$

and finally

$$\frac{n}{n-1} M \sigma_y^2 (1 - \sigma_y)^2 \lesssim \| \mathbf{J}_y \|_2^2 \lesssim 2 M \sigma_y^2 (1 - \sigma_y)^2 ,$$

or, in terms of the cross-entropy loss $l = -\log \sigma_y$:

$$\sqrt{\frac{nM}{n-1}} \mathrm{e}^{-l} \left( 1 - \mathrm{e}^{-l} \right) \lesssim \| \mathbf{J}_y \|_2 \lesssim \sqrt{2M} \mathrm{e}^{-l} \left( 1 - \mathrm{e}^{-l} \right). \tag{5}$$

We validate these approximate bounds in Figure App.11 (top).

**Full Jacobian**. Equation 5 establishes a close relationship between $\mathbf{J}_y$ and loss $l = -\log \sigma_y$, but of course, at test time we do not know the target class $y$. This allows us to only bound the full Jacobian norm from below:

$$\sqrt{\frac{nM}{n-1}} \mathrm{e}^{-l} \left( 1 - \mathrm{e}^{-l} \right) \lesssim \| \mathbf{J}_y \|_2 \leqslant \| \mathbf{J} \|_F . \tag{6}$$

For the upper bound, we assume the maximum-entropy case of $\sigma_y$: $\sigma_i \approx (1 - \sigma_y)/(n - 1)$, for $i \neq y$. The Jacobian norm is

$$\| \mathbf{J} \|_F^2 = \sum_{i=1}^n \| \mathbf{J}_i \|_2^2 = \| \mathbf{J}_y \|_2^2 + \sum_{i \neq y}^n \| \mathbf{J}_i \|_2^2 ,$$

where the first summand becomes:

$$\| \mathbf{J}_y \|_2^2 \approx M \sigma_y^2 \left[ (1 - \sigma_y)^2 + (n - 1) \left( \frac{1 - \sigma_y}{n - 1} \right)^2 \right] = \frac{Mn}{n-1} \sigma_y^2 (1 - \sigma_y)^2 ,$$

and each of the others

$$\| \mathbf{J}_i \|_2^2 \approx M \left( \frac{1 - \sigma_y}{n - 1} \right)^2 \left[ \left( 1 - \frac{1 - \sigma_y}{n - 1} \right)^2 + \left( \sigma_y^2 + (n - 2) \left( \frac{1 - \sigma_y}{n - 1} \right)^2 \right) \right]$$

$$= \frac{M}{(n-1)^3} (1 - \sigma_y)^2 \left( n \sigma_y^2 + n - 2 \right)^2 .$$

Adding $n - 1$ of such summands to $\| \mathbf{J}_y \|_2^2$ results in

$$\| \mathbf{J} \|_F \approx \frac{\sqrt{M}}{(n-1)} (1 - \sigma_y) \sqrt{n^2 \sigma_y^2 + n - 2} = \frac{\sqrt{M}}{(n-1)} \left( 1 - \mathrm{e}^{-l} \right) \sqrt{n^2 \mathrm{e}^{-2l} + n - 2}, \tag{7}$$

compared against the lower bound (Equation 6) and experimental data in Figure App.11.

---

[3]In the limit of infinite width, and fully Bayesian training, deep network logits are distributed exactly according to a Gaussian process (Neal, 1994; Lee et al., 2018). Similarly, each entry in the logit Jacobian also corresponds to an independent draw from a Gaussian process (Solak et al., 2003). It is therefore plausible that the Jacobian norm, consisting of a sum over the square of independent Gaussian samples in the correct limits, will tend towards a constant.

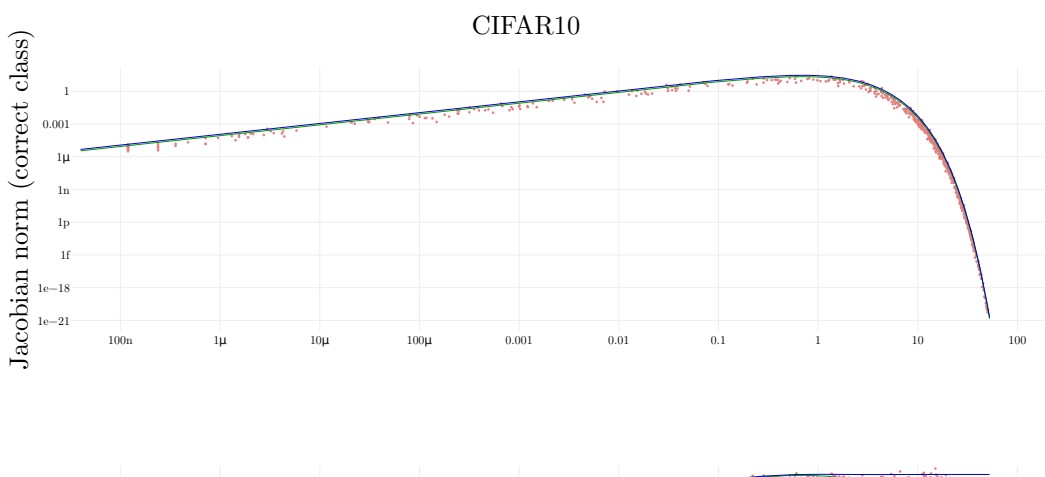

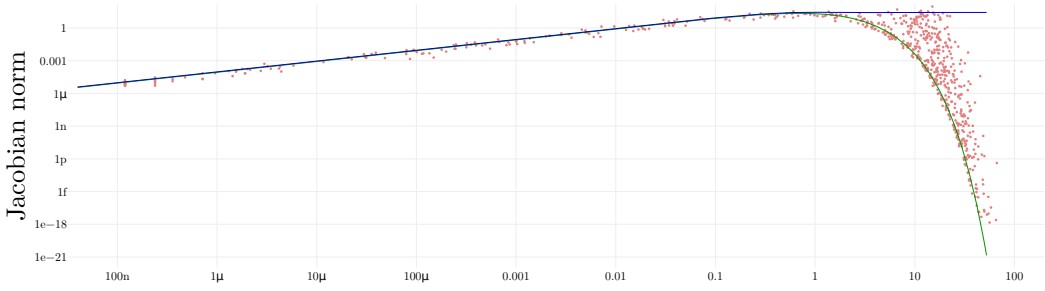

Cross-entropy loss

Figure App.11: **Top**: Jacobian norm $\left\|\mathbf{J}_y\left(\mathbf{x}\right)\right\|_2 = \left\|\partial\mathbf{f}_\sigma\left(\mathbf{x}\right)_y/\partial\mathbf{x}^T\right\|_2$ of the true class $y$ output probability is tightly related to the cross-entropy loss. Each point corresponds to one of the 1000 test inputs to a 100% trained network on CIFAR10, while lines depict analytic bounds from Equation 5. **Bottom**: Same experiment plotting the full Jacobian norm $\left\|\mathbf{J}\right\|_F$ against cross-entropy. Solid lines correspond to the lower bound from Equation 6 and the norm approximation from Equation 7. See §A.5.7 for experimental details and Figures 6 and App.9 for empirical evaluation of this relationship on multiple datasets and models.

## A.4   NON-LINEARITIES DEFINITIONS

Following activation functions are used in this work:

1. ReLU (Nair & Hinton, 2010): $\max(x, 0)$;
2. ReLU6 (Krizhevsky, 2010): $\min\left(\max(x, 0), 6\right)$;
3. Tanh: hyperbolic tangent, $(e^x - e^{-x})/(e^x + e^{-x})$;
4. HardTanh (Gulcehre et al., 2016): $\min\left(\max(x, -1), 1\right)$;
5. HardSigmoid (Gulcehre et al., 2016): $\min\left(\max(x + 0.5, 0), 1\right)$;

## A.5   EXPERIMENTAL SETUP

All experiments were implemented in Tensorflow (Abadi et al., 2016) and executed with the help of Vizier (Golovin et al., 2017). All networks were trained with cross-entropy loss. All networks were trained without biases. All computations were done with 32-bit precision. Learning rate decayed by a factor of 0.1 every 500 epochs.

Unless specified otherwise, initial weights were drawn from a normal distribution with zero mean and variance $2/n$ for ReLU, ReLU6 and HardSigmoid; $1/n$ for Tanh and HardTanh, where $n$ is the number of inputs to the current layer.

All inputs were normalized to have zero mean and unit variance, or, in other terms, lie on the $d$-dimensional sphere of radius $\sqrt{d}$, where $d$ is the dimensionality of the input.

All reported values, when applicable, were evaluated on the whole training and test sets of sizes $50000$ and $10000$ respectively. E.g. "generalization gap" is defined as the difference between train and test accuracies evaluated on the whole train and test sets.

When applicable, all trajectories/surfaces in input space were sampled with $2^{20}$ points.

### A.5.1 Plots and Error Bars

All figures except for 6 and App.11 are plotted with (pale) error bars (when applicable). The reported quantity was usually evaluated $8$ times with random seeds from $1$ to $8$[4], unless specified otherwise. E.g. if a network is said to be 100%-accurate on the training set, it means that each of the 8 randomly-initialized networks is 100%-accurate after training.

The error bar is centered at the mean value of the quantity and spans the standard error of the mean in each direction. If the bar appears to not be visible, it may be smaller than the mean value marker.

Weight initialization, training set shuffling, data augmentation, picking anchor points of data-fitted trajectories, selecting axes of a zero-centered elliptic trajectory depend on the random seed.

### A.5.2 Sensitivity along a Trajectory

Relevant figure 2.

A 20-layer ReLU-network of width 200 was trained on MNIST 128 times, with plots displaying the averaged values.

A random zero-centered ellipse was obtained by generating two axis vectors with normally-distributed entries of zero mean and unit variance (as such making points on the trajectory have an expected norm equal to that of training data) and sampling points on the ellipse with given axes.

A random data-fitted ellipse was generated by projecting three arbitrary input points onto a plane where they fall into vertices of an equilateral triangle, and then projecting their circumcircle back into the input space.

### A.5.3 Linear Region Boundaries

Relevant figure 3.

A 15-layer ReLU6-network of width 300 was trained on MNIST for $2^{18}$ steps using SGD with momentum (Rumelhart et al., 1988); images were randomly translated with wrapping by up to 4 pixels in each direction, horizontally and vertically, as well as randomly flipped along each axis, and randomly rotated by 90 degrees clockwise and counter-clockwise.

The sampling grid in input space was obtain by projecting three arbitrary input points into a plane as described in §A.5.2 such that the resulting triangle was centered at 0 and it's vertices were at a distance $0.8$ form the origin. Then, a sampling grid of points in the $[-1; 1]^{\times 2}$ square was projected back into the input space.

### A.5.4 Small Experiment

Relevant figures: 4 (second row) and 5 (bottom).

All networks were trained for $2^{18}$ steps of batch size of 256 using SGD with momentum. Learning rate was set to $0.005$ and momentum term coefficient to $0.9$.

Data augmentation consisted of random translation of the input by up to 4 pixels in each direction with wrapping, horizontally and vertically. The input was also flipped horizontally with probability $0.5$. When applying data augmentation (second row of Figure 4), the network is unlikely to

---

[4]If a particular random seed did not finish, it was not taken into account; we believe this nuance did not influence the conclusions of this paper.

encounter the canonical training data, hence few configurations achieved $100\%$ training accuracy. However, we verified that all networks trained with data augmentation reached a higher test accuracy than their analogues without, ensuring that the generalization gap shrinks not simply because of lower training accuracy.

For each dataset, networks of width $\{100, 200, 500, 1000, 2000, 3000\}$, depth $\{2, 3, 5, 10, 15, 20\}$ and activation function $\{\text{ReLU, ReLU6, HardTanh, HardSigmoid}\}$ were evaluated on $8$ random seeds from $1$ to $8$.

### A.5.5  LARGE EXPERIMENT

Relevant figures: 1, 4 (except for the second row), 5 (top), App.8.

335671 networks were trained for $2^{19}$ steps with random hyper-parameters; if training did not complete, a checkpoint at step $2^{18}$ was used instead, if available. When using L-BFGS, the maximum number of iterations was set to 2684. The space of available hyper-parameters included[5]:

1. CIFAR10 and CIFAR100 datasets cropped to a $24 \times 24$ center region;
2. all 5 non-linearities from §A.4;
3. SGD, Momentum, ADAM (Kingma & Ba, 2014), RMSProp (Hinton et al., 2012) and L-BFGS optimizers;
4. learning rates from $\{0.01, 0.005, 0.0005\}$, when applicable. Secondary coefficients were fixed at default values of Tensorflow implementations of respective optimizers;
5. batch sizes of $\{128, 512\}$ (unless using L-BFGS with the full batch of 50000);
6. standard deviations of initial weights from $\{0.5, 1, 4, 8\}$ multiplied by the default value described in §A.5;
7. widths from $\left\{1, 2, 4, \cdots, 2^{16}\right\}$;
8. depths from $\left\{2, 3, 5, \cdots, 2^{6} + 1\right\}$;
9. true and random training labels;
10. random seeds from 1 to 8.

### A.5.6  PER-POINT GENERALIZATION

Relevant figures 6, App.9.

Networks were with either cross-entropy or $\ell_2$-loss trained for $2^{19}$ steps on whole datasets (CIFAR100, CIFAR10, Fashion-MNIST and MNIST) and evaluated on random subsets of 1000 test images.

Hyper-parameters were: non-linearity (all functions from §A.4), width (50, 100, 200, 500, 1000), depth (2, 5, 10, 20, 30), learning rate (0.0001, 0.001, 0.01), optimizer (SGD, Momentum, ADAM, RMSProp). Only one random seed (1) was used. For each dataset a random subset of 5 configurations among all the $100\%$-accurate (on training) networks was plotted (apart from the case of CIFAR100, where networks of training accuracy of at least $99.98\%$ were selected).

### A.5.7  CROSS-ENTROPY AND SENSITIVITY ANALYSIS

Relevant figure App.11.

Networks were trained for $2^{18}$ on the whole CIFAR10 training set and evaluated networks on a random test subset of size 1000. The hyper-parameters consisted of non-linearity (all functions from §A.4), width (50, 100 or 200) and depth (2, 5, 10, 20). Only one random seed (1) was considered. A single random $100\%$-accurate (on training data) network was drawn to compare experimental measurements with analytic bounds on the Jacobian norm.

---

[5]Due to time and compute limitations, this experiment was set up such that configurations of small size were more likely to get evaluated (e.g. only a few networks of width $2^{16}$ were trained, and all of them had depth 2). However, based on our experience with smaller experiments (where each configuration got evaluated), we believe this did not bias the findings of this paper, while allowing them to be validated across a very wide range of scenarios.

