# OpenReview forum: "Sensitivity and Generalization in Neural Networks: an Empirical Study"
_ICLR.cc/2018/Conference — Accept (Poster)_

### Official Review · AnonReviewer2 · 2017-11-27
**The authors present a large scale empirical evaluation on sensitivity and generalization for DNNs within the scope of image classification. They convincingly present strong empirical evidence for the F-norm of the input-output Jacobian to be predictive/informative of generalization with large scale DNNs.**

**Rating:** 8
**Confidence:** 4

**Review:**

The authors have undertaken a large scale empirical evaluation on sensitivity and generalization for DNNs within the scope of image classification. They are investigating the suitability of the F-norm of the input-output Jacobian in large scale DNNs and they evaluate sensitivity and generalization metrics across the input space, both on and of the data manifold. They convincingly present strong empirical evidence for the F-norm of the Jacobian to be predictive and informative of generalization of the DNN within the image classification domain.

The paper is well written. The problem is clearly presented and motivated. Most potential questions of a reader as well as interesting details are supplied by footnotes and the appendix.
The contributions are to my knowledge both novel and significant.
The paper seem to be technically correct. The methodology and conclusions are reasonable.
I believe that this is important work and applaud the authors for undertaking it. I hope that the interesting leads will be further investigated and that similar studies will be conducted beyond the scope of image classification. '
The research and further investigations would be strengthened if they would include a survey on the networks presented in the literature in a similar manner as the authors did with the generated networks within the presented study. For example compare networks from benchmark competitions in terms of sensitivity and generalization using the metrics presented here.

Please define "generalization gap" and show how you calculate/estimate it. The term us used differently in much of the machine learning literature(?). Given this and that the usually sought after generalization error is unobtainable due to the unknown joint distribution over data and label, it is necessary to clarify the precise meaning of "generalization gap" and how you calculated it. I intuitively understand but I am not sure that the metric I have in mind is the same as the one you use. Such clarification will also improve the accessibility for a wider audience.

Figure 4:
I find Figure 4:Center a bit confusion. Is it there to show where on the x-axis of Figure 4:Top,the three points are located? Does this mean that the points are not located at pi/3, pi, 5pi/3 as indicated in the figure and the vertical lines of the figure grid? If it is not, then is it maybe possible to make the different sub-figure in Figure 4 more distinctive, as to not visually float into each other?

Figure 5:
The figure makes me curious about what the regions look like close to the training points, which is currently hidden by the content of the inset squares. Maybe the square content can be made fully transparent so that only the border is kept? The three inset squares could be shown right below each sub-figure, aligned at the x-axis with the respective position of each of the data points.

---

> ### Author Response · Authors · 2017-12-19
> **Response to AnonReviewer2**
>
>
> (1)
> >> Please define "generalization gap" and show how you calculate/estimate it. The term us used differently in much of the machine learning literature(?). Given this and that the usually sought after generalization error is unobtainable due to the unknown joint distribution over data and label, it is necessary to clarify the precise meaning of "generalization gap" and how you calculated it. I intuitively understand but I am not sure that the metric I have in mind is the same as the one you use. Such clarification will also improve the accessibility for a wider audience.
>
> Thank you for the comment, we have included the definition in the Appendix A.4 in the second revision.
>
> We define generalization gap as the difference between train and test accuracy on the whole train and test sets. Precisely,
>
> Generalization gap = (# correctly classified training images)/(50K) - (# correctly classified test images)/(10K).
>
> (all training and test sets are of size 50K and 10K respectively)
> ------------------------------------------------------------------------------------------------------------------------------------------------------
> (2)
> >> Figure 4:
> I find Figure 4:Center a bit confusion. Is it there to show where on the x-axis of Figure 4:Top,the three points are located? Does this mean that the points are not located at pi/3, pi, 5pi/3 as indicated in the figure and the vertical lines of the figure grid? If it is not, then is it maybe possible to make the different sub-figure in Figure 4 more distinctive, as to not visually float into each other?
>
> We apologize for the confusing figure. Central figure (now top in the second revision) is there for the exact reason you mention (to show where the points are located, which indeed should be pi/3, pi, 5pi/3). We have separated the subfigures further and aligned the digits with the values of pi/3, pi and 5pi/3 precisely in the second revision.
> ------------------------------------------------------------------------------------------------------------------------------------------------------
> (3)
> >> Figure 5:
> The figure makes me curious about what the regions look like close to the training points, which is currently hidden by the content of the inset squares. Maybe the square content can be made fully transparent so that only the border is kept? The three inset squares could be shown right below each sub-figure, aligned at the x-axis with the respective position of each of the data points.
>
> Thank you for the interesting suggestion! We have produced the requested figures with only the boundaries overlayed:
> -- before training: https://www.dropbox.com/s/14xcvoval4eluz4/boundaries_before_transparent.png?dl=1;
> -- after training: https://www.dropbox.com/s/lj7y9eimnqw0lsd/boundaries_after_transparent.png?dl=1.
>
> We do not observe any special behavior at such scale. This is in agreement with Figure 3 (bottom), showing that the density of transitions around the points changes slowly.
> ------------------------------------------------------------------------------------------------------------------------------------------------------
>
> Thank you for the detailed review and helpful comments! We are pleased that you found our work useful.

---

### Official Review · AnonReviewer3 · 2017-11-27
**Interesting experimental setting and validation metrics. Unclear presentation and modelling rationale**

**Rating:** 5
**Confidence:** 3

**Review:**

This work investigates sensitivity and generalisation properties of neural networks with respect to a number of metrics aimed at quantifying the robustness with respect to data variability, varying parameters and representativity of training/testing data.
The validation is based on the Jacobian of the network, and in the detection of the “transitions” associated to the data space. These measures are linked, as the former quantifies the sensitivity of the network respect to infinitesimal data variations, while the latter quantifies the complexity of the modelled data space.
The study explores a number of experimental setting, where the behaviour of the network is analysed on synthetic paths around training data, from pure random data points, to curves interpolating different/same data classes.
The experimental results are performed on CIFAR10,CIFAR100, and MNIST. Highly-parameterised networks seem to offer a better generalisation, while lower Jacobian norm are usually associated to better generalisation and fewer transitions, and can be obtained with data augmentation.

The paper proposes an interesting analysis aimed at the empirical exploration of neural network properties, the proposed metrics provide relevant insights to understand the behaviour of a network under varying data points.

Major remarks.

The proposed investigation is to my opinion quite controversial. Interesting data variation does not usually corresponds to linear data change. When considering the linear interpolation of training data, the authors are actually creating data instances not compatible with the original data source: for example, the pixel-wise intensity average of digits is not a digit anymore. For this reason, the conclusions drawn about the model sensitivity are to my opinion based a potentially uninteresting experimental context. Meaningful data variation can be way more complex and high-dimensional, for example by considering spatial warps of digits, or occlusions and superpositions of natural images. This kind of variability is likely to correspond to real data changes, and may lead to more reliable conclusions. For this reason, the proposed results may provide little indications of the true behaviour of the models data in case of meaningful  data variations.

Moreover, although performed within a cross-validation setting, training and testing are still applied to the same dataset. Cross-validation doesn’t rule out validation bias, while it is also known that the classification performance significantly drops when applied to independent “unseen” data, provided for example in different cohorts. I would expect that highly parameterised models would lead to worse performance when applied to genuinely independent cohorts, and I believe that this work should extend the investigation to this experimental setting.

Minor remarks.

The authors should revise the presentation of the proposed work. The 14 figures(!) of main text are not presented in the order of appearance. The main one (figure 1) is provided in the first paragraph of the introduction and never discussed in the rest of the paper.

---

> ### Author Response · Authors · 2017-12-19
> **Rebuttal to AnonReviewer3, part 2**
>
>
> (2)
> >> Moreover, although performed within a cross-validation setting, training and testing are still applied to the same dataset. Cross-validation doesn’t rule out validation bias, while it is also known that the classification performance significantly drops when applied to independent “unseen” data, provided for example in different cohorts. I would expect that highly parameterised models would lead to worse performance when applied to genuinely independent cohorts, and I believe that this work should extend the investigation to this experimental setting.
>
> We would like to address your concern. Could you please expand on what you mean by "genuinely independent cohorts"?
>
> Are you concerned that MNIST images in train and test may have digits written by same individuals? If so, we believe this should be less of a problem in Fashion-MNIST, CIFAR10, and CIFAR100 datasets where we see similar results.
>
> We would like to provide some additional information regarding our training and evaluation procedure, hoping that this might address your concern. Train and test data are balanced random 50K and 10K i.i.d. samples respectively. We train all our networks for a large number of gradient steps (2^18 or 2^19 when applicable) without any regularization / validation / early stopping. We then evaluate all quantities mentioned in the paper on the whole 50K or 10K datasets respectively, when applicable.
>
> Please let us know if the above answers your question. If no, we will be happy to expand on it once we fully understand your request.
> ------------------------------------------------------------------------------------------------------------------------------------------------------
> (3)
> >> The authors should revise the presentation of the proposed work. The 14 figures(!) of main text are not presented in the order of appearance.
>
> Thank you, we have rearranged the figures in the order of presentation in the second revision.
> ------------------------------------------------------------------------------------------------------------------------------------------------------
> (4)
> >> The main one (figure 1) is provided in the first paragraph of the introduction and never discussed in the rest of the paper.
>
> While we acknowledge the importance of this figure, we consider it as motivation for our study (ultimately leading to key results presented in figures 3 [sensitivity along a trajectory intersecting the data manifold] and 4 in the first paper revision / 9 in the second revision [Jacobian norm correlating with generalization]), which is why it is only mentioned in the introduction.
> ------------------------------------------------------------------------------------------------------------------------------------------------------
>
> We thank you for the careful and insightful review. We believe that we were able to address your concerns both in terms of rebuttal and new experimental evidence, and hope that you will raise your score as a result.

---

> ### Author Response · Authors · 2017-12-19
> **Rebuttal to AnonReviewer3, part 1**
>
>
> (1)
> >> The proposed investigation is to my opinion quite controversial. Interesting data variation does not usually corresponds to linear data change. When considering the linear interpolation of training data, the authors are actually creating data instances not compatible with the original data source: for example, the pixel-wise intensity average of digits is not a digit anymore. For this reason, the conclusions drawn about the model sensitivity are to my opinion based a potentially uninteresting experimental context. Meaningful data variation can be way more complex and high-dimensional, for example by considering spatial warps of digits, or occlusions and superpositions of natural images. This kind of variability is likely to correspond to real data changes, and may lead to more reliable conclusions. For this reason, the proposed results may provide little indications of the true behaviour of the models data in case of meaningful  data variations.
>
> We believe there to be two potential concerns in this remark and shall address them separately.
>
>
> 1.A) The concern of linear interpolation of the training data being incompatible with the original data source, and as such our sampling trajectories in section 4.1 not containing meaningful data variations. This is by design! Such a trajectory will indeed lie mostly outside of the data manifold, yet intersect it in 3 points. Measuring our metrics along these trajectories allow us to draw conclusions about the behavior of a trained neural network near the data manifold and away from it (Figure 3).
>
> Otherwise, in the rest of the paper, transitions are counted along a trajectory interpolating horizontal translations of an image (see definition in section 3.2), which do represent a complex curve along a meaningful data variation (translation). We agree that analyzing a richer set of transformations within the data manifold would be interesting. However, characterizing data variation is a complex field of study, and we believe that translations provide a well defined and tractable set of transformations which typically remain within the data distribution.
>
> Finally, the best-performing metric of our work, that is the Frobenius norm of the Jacobian (see definition in section 3.1) is averaged over individual data points and does not hinge on any kind of interpolation!
>
>
> 1.B)  We can alternatively interpret your remark as being skeptical regarding the Jacobian norm reflecting sensitivity with respect to meaningful data variations. Indeed, it does not, and as described in section 3.1, it reflects sensitivity to isotropic perturbations.
>
> Motivated by this concern, we have performed an additional experiment measuring the norm of the Jacobian of the output with respect to horizontal shift, hence sensitivity to a meaningful data variation (translation) along the data manifold (bottom part of the figure, in contrast to the top): https://www.dropbox.com/s/cmh2s3eqb7vihj9/horizontal_translation_jacobian.pdf?dl=1
>
> We observe an effect qualitatively similar to (yet less noisy than) when the Frobenius norm of the input-output Jacobian is considered (top part of the figure, or Figure 8 in the first paper revision / Figure  9 (bottom) in the second revision).
>
> However, we believe our current results still provide a useful insight. Different datasets have different axes of data variations. Which those are may not always be clear: indeed, understanding all the meaningful axes of data variations would essentially amount to solving the problem of generating natural images. Yet in the absence of preconceived notions of what directions are meaningful, the input-output Jacobian can be a universal metric indicative of generalization, as evidenced by our experiments on 4 datasets (MNIST, Fashion-MNIST, CIFAR10, CIFAR100) which definitely have different notions of meaningful data variations.
> ------------------------------------------------------------------------------------------------------------------------------------------------------

---

> > ### Comment · AnonReviewer3 · 2018-01-19
> > **Not very insightful experimental design**
> >
> > I thank the authors for the detailed reply to my comments.
> > I may understand the idea of studying the model behaviour on data laying far from the training data. However, I believe that the proposed experimental paradigm still does not allow to draw insightful conclusions. Indeed, it is clearly expected that the performance of the networks sensibly drops in point far away from the data support. Similarly, spatial translations too present data compatibility issues, as they lead to non representative data points ( such as the cropped digits shown in Figure 2).
> >
> > To my opinion, it could have been more useful to investigate the model performance in case of meaningful data variation, representing indeed "hard" and realistic testing cases. As I pointed in my previous review, these cases may be represented by nonlinear spatial warping: generating such nonlinear paths in the manifold would be significantly more meaningful.

---

> > > ### Author Response · Authors · 2018-01-22
> > > **Response to the update of AnonReviewer3**
> > >
> > > Thank you for taking the time to consider and respond to our rebuttal!
> > > ------------------------------------------------------------------------------------------------------------------------------------------------------
> > >
> > > (1)
> > > >> Indeed, it is clearly expected that the performance of the networks sensibly drops in point far away from the data support.
> > >
> > > This statement is true, but we strongly disagree that it detracts from our paper:
> > > 1.1) This is not the statement we are making / evaluating in the paper (yet it is implied as motivation for experiments in section 4.1; see comments below);
> > > 1.2) Our findings in section 4.1 (Figure 3), is that the _sensitivity_ of the network to small input perturbations increases away from the data. To the best of our knowledge, this result is novel and not obvious.
> > > 1.3) This common-sense expectation only corroborates the correlation we establish between sensitivity and generalization (i.e. “trained networks perform poorly away from training data” and “trained networks are highly sensitive away from training data”) in other numerous experiments.
> > > ------------------------------------------------------------------------------------------------------------------------------------------------------
> > >
> > > (2)
> > > >>Similarly, spatial translations too present data compatibility issues, as they lead to non representative data points ( such as the cropped digits shown in Figure 2).
> > >
> > > >>To my opinion, it could have been more useful to investigate the model performance in case of meaningful data variation, representing indeed "hard" and realistic testing cases. As I pointed in my previous review, these cases may be represented by nonlinear spatial warping: generating such nonlinear paths in the manifold would be significantly more meaningful.
> > >
> > > We strongly disagree with this argument and hope that you will reconsider this perspective in light of the key points below:
> > > 2.1) As we stated in the paper and rebuttal, the best-performing metric in our work (Frobenius norm of the Jacobian) is evaluated at individual data points and _has nothing to do at all_ with the trajectories used to evaluate the other metric (linear region transitions). As such this concern cannot apply to the relationship between Jacobian norm and generalization.
> > >
> > > 2.2) The statement that other spatial warpings are more relevant/meaningful than horizontal translation on the datasets we consider (MNIST, Fashion-MNIST, CIFAR10, CIFAR100) requires justification. Without appropriate data augmentation, spatial warping will generate images that are not representative of the true data distribution as well. We emphasize that the trajectory interpolating horizontal translations is _not a linear trajectory in input space_ (see Figure 2) and we see no obvious reasons to assume it is simpler than other warping transformations.
> > > ------------------------------------------------------------------------------------------------------------------------------------------------------
> > >
> > > Thank you again for carefully considering our paper.

---

### Official Review · AnonReviewer1 · 2017-11-28
**The paper is poorly organized, the presented analysis is hard to read, the obtained results (and mainly all figures) are unclear.**

**Rating:** 4
**Confidence:** 5

**Review:**

This paper proposes an analysis of the robustness of deep neural networks with respect to data perturbations.

*Quality*
The quality of exposition is not satisfactory. Actually, the paper is pretty difficult to evaluate at the present stage and it needs a drastic change in the writing style.

*Clarity*
The paper is not clear and highly unstructured.

*Originality*
The originality is limited for what regards Section 3: the proposed metrics are quite standard tools from differential geometry. Also, the idea of taking into account the data manifold is not brand new since already proposed in “Universal Adversarial Perturbation” at CVPR 2017.

*Significance*
Due to some flaws in the experimental settings, the relevance of the presented results is very limited. First, the authors essentially exploit a customized architecture, which has been broadly fine-tuned regarding hyper-parameters, gating functions and optimizers. Why not using well established architectures (such as DenseNets, ResNets, VGG, AlexNet)?
Moreover, despite having a complete portrait of the fine-tuning process is appreciable, this compromises the clarity of the figures which are pretty hard to interpret and absolutely not self-explanatory: probably it’s better to only consider the best configuration as opposed to all the possible ones.
Second, authors assume that circular interpolation is a viable way to traverse the data manifold. The reviewer believes that it is an over-simplistic assumption. In fact, it is not guaranteed a priori that such trajectories are geodesic curves so, a priori, it is not clear why this could be a sound technique to explore the data manifold.

CONS:
The paper is difficult to read and needs to be thoroughly re-organized. The problem is not stated in a clear manner, and paper’s contribution is not outlined. The proposed architectures should be explained in detail. The results of the sensitivity analysis should be discussed in detail. The authors should explain the approach of traversing the data manifold with ellipses (although the reviewer believes that such approach needs to be changed with something more principled). Figures and results are not clear.
The authors are kindly asked to shape their paper to match the suggested format of 8 pages + 1 of references (or similar). The work is definitely too long considered its quality. Additional plots and discussion can be moved to an appendix.
Despite the additional explanation in Footnote 6, the graphs are not clear. Probably authors should avoid to present the result for each possible configuration of the hyper-parameters, gatings and optimizers and just choose the best setting.
Apart from the customized architecture, authors should have considered established deep nets, such as DenseNets, ResNets, VGG, AlexNet.
The idea of considering the data manifold within the measurement of complexity is a nice claim, which unfortunately is paired with a not convincing experimental analysis. Why ellipses should be a proper way to explore the data manifold? In general, circular interpolation is not guaranteed to be geodesic curves which lie on the data manifold.

Minor Comments:
Sentence to rephrase: “We study common in the machine learning community ways to ...”
Please, put the footnotes in the corresponding page in which it is referred.
The reference to ReLU is trivially wrong and need to be changed with [Nair & Hinton ICML 2010]

**UPDATED EVALUATION AFTER AUTHORS' REBUTTAL**
We appreciated the effort in providing specific responses and we also inspected the updated version of the paper. Unfortunately, despite the authors' effort, the reviewer deems that the conceptual issues that have been highlighted are still present in the paper which, therefore, is not ready for acceptance yet.

---

> ### Author Response · Authors · 2017-12-19
> **Rebuttal to AnonReviewer1, part 2**
>
>
> (5)
> >> Second, authors assume that circular interpolation is a viable way to traverse the data manifold. The reviewer believes that it is an over-simplistic assumption. In fact, it is not guaranteed a priori that such trajectories are geodesic curves so, a priori, it is not clear why this could be a sound technique to explore the data manifold.
>
> The interpretation of the reviewer is incorrect. Nowhere in the text did we claim to traverse the data manifold with an ellipse.
>
> Ellipses intersecting the data manifold at 3 specific points are studied only in section 4.1 to track our metrics along a trajectory with respect to how close a point to the data manifold is, which is an appropriate choice for this purpose.
>
> In the rest of the paper, the transitions metric is computed along a trajectory that interpolates horizontal translations of an image (as stated in section 3.2), which is not an ellipse and generally lies within the data manifold for translation invariant datasets. In fact we augment the training data with translations for one experiment.
>
> We have edited the toy illustration of such a trajectory in Figure 2 (right) to make this more clear in the second revision.
>
> In addition, we never claimed any of our curves to be geodesics and fail to see how this property is necessary for the purpose of our work.
>
> Finally, the relationship between the Frobenius norm of the network Jacobian and generalization is averaged over individual data points and has nothing to do with traversing the data manifold.
>
> We thank the reviewer for the useful feedback and will improve the exposition in the next revision.
> ------------------------------------------------------------------------------------------------------------------------------------------------------
> (6)
> >> The authors should explain the approach of traversing the data manifold with ellipses (although the reviewer believes that such approach needs to be changed with something more principled).
>
> Please see discussion about trajectories above (5).
> ------------------------------------------------------------------------------------------------------------------------------------------------------
> (7)
> >> Probably authors should avoid to present the result for each possible configuration of the hyper-parameters, gatings and optimizers and just choose the best setting.
>
> Please see the relevant discussion above (4).
> ------------------------------------------------------------------------------------------------------------------------------------------------------
> (8)
> >> Apart from the customized architecture, authors should have considered established deep nets, such as DenseNets, ResNets, VGG, AlexNet.
>
> Please see relevant discussion above (3).
> ------------------------------------------------------------------------------------------------------------------------------------------------------
> (9)
> >> The idea of considering the data manifold within the measurement of complexity is a nice claim, which unfortunately is paired with a not convincing experimental analysis. Why ellipses should be a proper way to explore the data manifold? In general, circular interpolation is not guaranteed to be geodesic curves which lie on the data manifold.
>
> Please see the relevant discussion above (5).
> ------------------------------------------------------------------------------------------------------------------------------------------------------
> (10)
> >> Sentence to rephrase: “We study common in the machine learning community ways to ...”
>
> Thank you, we have changed the wording in the second revision.
> ------------------------------------------------------------------------------------------------------------------------------------------------------
> (11)
> >> Please, put the footnotes in the corresponding page in which it is referred.
>
> Thank you, we have fixed the footnotes in the second revision.
> ------------------------------------------------------------------------------------------------------------------------------------------------------
> (12)
> >> The reference to ReLU is trivially wrong and need to be changed with [Nair & Hinton ICML 2010]
>
> Thank you for pointing this out, we have fixed the reference in the second revision.
> ------------------------------------------------------------------------------------------------------------------------------------------------------
>
> We thank the reviewer for the detailed feedback! We will work to improve the clarity of our work in the next revision.

---

> ### Author Response · Authors · 2017-12-19
> **Rebuttal to AnonReviewer1, part 1**
>
>
> (1)
> >> The originality is limited for what regards Section 3: the proposed metrics are quite standard tools from differential geometry.
>
> We did not claim to propose novel metrics anywhere in the paper; on the contrary, we cite prior work that used / introduced them in section 2.
>
> The novelty of this work is in performing extensive evaluation of these metrics on trained neural networks and relating them to generalization (which is also emphasized multiple times in section 2).
> ------------------------------------------------------------------------------------------------------------------------------------------------------
> (2)
> >> Also, the idea of taking into account the data manifold is not brand new since already proposed in “Universal Adversarial Perturbation” at CVPR 2017.
>
> Thank you for the very interesting reference! We now cite it in related work in the second revision.
>
> However, we disagree with the claim that this work diminishes the novelty of ours. Nowhere in the paper did we assert to be the first to "take into account the data manifold". Our claim was to compare behavior of a trained network on and off the data manifold (see last paragraph of section 2).
>
> A great deal of previous research has examined the statistics of natural stimuli (e.g. dating at least as far back as Barlow, 1959). The specific paper you reference is a very interesting exploration of universal adversarial perturbations. However, it does not investigate the behavior of the network outside of the data manifold.
> ------------------------------------------------------------------------------------------------------------------------------------------------------
> (3)
> >> Due to some flaws in the experimental settings, the relevance of the presented results is very limited. First, the authors essentially exploit a customized architecture, which has been broadly fine-tuned regarding hyper-parameters, gating functions and optimizers. Why not using well established architectures (such as DenseNets, ResNets, VGG, AlexNet)?
>
> Thank you for your suggestion! Evaluating our metrics on the proposed architectures is indeed a very interesting direction for future research.
>
> However, we disagree that this establishes a flaw in our experiments. On the contrary, the architectures you suggest are ones that are extremely customized and fine-tuned, while the set of fully-connected (FC) architectures we consider are quite generic.
>
> The reason for considering FC networks in this work was to perform a large-scale evaluation of the computationally-intensive metrics in a very wide variety of settings, to understand the resulting distribution over network behaviors, rather than measuring the behavior in a small number of hand-tuned scenarios (while extending the hundreds of thousands of experiments performed in this work onto complex convolutional architectures is beyond the scope of this work).
>
> We further emphasize that:
> -- Almost all networks considered in this work have achieved 100% accuracy on the whole training set.
> -- Best-performing configurations yield test accuracies competitive with state-of-the-art results for FC networks.
> -- We evaluate our results on 4 different datasets of varying complexity (MNIST, Fashion-MNIST, CIFAR10, CIFAR100).
> For this reason we believe our work presents results that are both comprehensive and appropriate to different generalization regimes.
>
> We will make sure to improve our presentation and emphasize the above in our next revision.
> ------------------------------------------------------------------------------------------------------------------------------------------------------
> (4)
> >> Moreover, despite having a complete portrait of the fine-tuning process is appreciable, this compromises the clarity of the figures which are pretty hard to interpret and absolutely not self-explanatory: probably it’s better to only consider the best configuration as opposed to all the possible ones.
>
> This change would run counter to a principal strength of the paper -- that we examine the distribution over network behavior for a wide range of hyper-parameters and datasets (over thousands of experiments). Showing only the best-performing models would completely obfuscate the insights drawn from this analysis and significantly detract from our paper.
>
> We will make sure to revise our presentation to make this point more clear.
> ------------------------------------------------------------------------------------------------------------------------------------------------------

---

> ### Author Response · Authors · 2018-01-05
> **Response to the Updated Evaluation of AnonReviewer1**
>
> Thank you for the updated evaluation and for taking the time to review the revised submission.
>
> We believe we have addressed all of the original concerns in our rebuttal. We would greatly appreciate any further feedback on what specific "conceptual issues" remain unresolved, so that we can use your feedback to further improve our paper.
>
> We have also uploaded a new revision that further improves clarity and exposition, especially related to areas that you flagged as unclear.

---

> > ### Comment · AnonReviewer1 · 2018-01-20
> > **Response to authors' response**
> >
> > We appreciated the effort provided by the authors in improving the paper. However the following problems still remain.
> >
> > 1. The authors did not include well established architectures (DenseNets, VGG, AlexNet) for the experiments.
> >
> > 2. Presenting the results for any possible parameter configuration makes the plot pretty hard to read.
> >
> > 3. The approach of elliptic interpolation still remains not fully convincing - as AnonReviewer3 also pointed out.
> >
> > 4. The paper is not yet compliant with the suggested format and the added material does not fully justify the exceeded length.

---

### Public Comment · ~Tom_Zahavy1 · 2018-03-26
**Interesting investigation of the robustness of neural networks**

Congratulations on the acceptance of this impressive work.

Similar to the authors, we believe that a large-scale statistical study of deep neural networks can provide new insights on the connection between generalization and robustness of deep neural networks.

The authors mention in their paper that "They derive bounds for the generalization gap in terms of the Jacobian norm within the framework of algorithmic robustness (Xu & Mannor, 2012). Our results provide empirical support for their conclusions through an extensive number of experiments".

We wish to point out the authors to our paper "Ensemble Robustness And Generalization of Stochastic Deep Learning Algorithms", that originally appeared in 2016 (https://arxiv.org/abs/1602.02389) while the latest version was submitted to ICLR18 (concurrently to this work) and got accepted to the workshop track (https://openreview.net/pdf?id=HyJf8QJDz).

Our paper extended the work in (Xu & Mannor, 2012) to the case of stochastic algorithms (e.g., sgd and dropout), that are common in the training of deep neural networks. We supported these theoretical results with an empirical evaluation that is very similar in spirit to the one performed by the authors.

We find these works to be closely related and suggest discussing the similarities in the text.

---

> ### Author Response · Authors · 2018-04-13
> **Thank you!**
>
> Thank you for the interesting reference! We will mention it in related work in the next arXiv revision.

---

### Decision · Program_Chairs · 2018-01-29
**ICLR 2018 Conference Acceptance Decision**

**Decision:**

Accept (Poster)

**Comment:**

Reviewers always find problems in papers like this.

AnonReviewer1 would have preferred to have seen a study of traditional architectures, rather than fully connected ones, which are now less frequently used. They thought the paper was too long, the figures too cluttered, and were not convinced by the discussion around linear v. elliptical trajectories.

I appreciate the need for a parametrizable architecture, although it may not be justified to translate these insights to other architectures, and then the fact that fully connected architectures are less common undermines the impact of the work. I don't find the length a problem, and I don't find the figures a problem.

After the back and forth, AnonReviewer3 believes that there are data compatibility issues associated with the studied transformations and that non-linear transformations would have been more informative. I find the reviewers response to be convincing.

AnonReviewer2 is strongly in favor of acceptance, finding the work exhaustive, interesting, and of high quality. I'm inclined to agree.

---

> ### Public Comment · (anonymous) · 2018-02-02
> **What do you mean exactly by "like this"?**
>
> Reviewers find problems in any, since no paper is perfect.

---

> > ### Public Comment · (anonymous) · 2018-03-19
> > **Checkout my paper**
> >
> > My paper has no mistakes. Check it out!